

# Evolutionary origins of the emergent ST796 clone of vancomycin resistant *Enterococcus faecium*

Andrew H. Buultjens[1], Margaret M.C. Lam[1], Susan Ballard[2], Ian R. Monk[1], Andrew A. Mahony[3], Elizabeth A. Grabsch[3], M. Lindsay Grayson[3], Stanley Pang[4,5], Geoffrey W. Coombs[4,5], J. Owen Robinson[4,6], Torsten Seemann[7], Paul D.R. Johnson[3,8], Benjamin P. Howden[2] and Timothy P. Stinear[1]

[1] Department of Microbiology and Immunology, Doherty Institute for Infection and Immunity, University of Melbourne, Melbourne, Victoria, Australia
[2] Microbiology Diagnostic Unit, Department of Microbiology and Immunology, Doherty Institute for Infection and Immunity, University of Melbourne, Melbourne, Victoria, Australia
[3] Infectious Diseases Department, Austin Health, Heidelberg, Victoria, Australia
[4] School of Veterinary and Life Sciences, Murdoch University, Murdoch, Western Australia, Australia
[5] Department of Microbiology, Pathwest Laboratory Medicine-WA, Fiona Stanley Hospital, Murdoch, Western Australia, Australia
[6] Department of Infectious Diseases, Fiona Stanley Hospital, Murdoch, Western Australia, Australia
[7] Victorian Life Sciences Computation Initiative, University of Melbourne, Carlton, Victoria, Australia
[8] Department of Medicine, University of Melbourne, Heidelberg, Victoria, Australia

Corresponding authors
Paul D.R. Johnson,
Paul.Johnson@austin.org.au
Timothy P. Stinear,
tstinear@unimelb.edu.au

## ABSTRACT

From early 2012, a novel clone of vancomycin resistant *Enterococcus faecium* (assigned the multi locus sequence type ST796) was simultaneously isolated from geographically separate hospitals in south eastern Australia and New Zealand. Here we describe the complete genome sequence of Ef_aus0233, a representative ST796 *E. faecium* isolate. We used PacBio single molecule real-time sequencing to establish a high quality, fully assembled genome comprising a circular chromosome of 2,888,087 bp and five plasmids. Comparison of Ef_aus0233 to other *E. faecium* genomes shows Ef_aus0233 is a member of the epidemic hospital-adapted lineage and has evolved from an ST555-like ancestral progenitor by the accumulation or modification of five mosaic plasmids and five putative prophage, acquisition of two cryptic genomic islands, accrued chromosomal single nucleotide polymorphisms and a 80 kb region of recombination, also gaining Tn*1549* and Tn*916*, transposons conferring resistance to vancomycin and tetracycline respectively. The genomic dissection of this new clone presented here underscores the propensity of the hospital *E. faecium* lineage to change, presumably in response to the specific conditions of hospital and healthcare environments.

## INTRODUCTION

*Enterococcus faecium* is a human and animal gastrointestinal tract (GIT) commensal but a lineage within the species has rapidly evolved to become a significant opportunistic pathogen (*Coombs et al., 2014*; *Deshpande et al., 2007*).

Early genotyping methods such as amplified fragment length polymorphism (*Willems et al., 2000*), restriction endonuclease analysis (*Quednau, Ahrne & Molin, 1999*), multi-locus sequence typing (MLST) (*Homan et al., 2002*; *Leavis, Bonten & Willems, 2006*; *Top, Willems & Bonten, 2008*) and more recent analyses using whole genome datasets (*Lebreton et al., 2013*), have shown the *E. faecium* population separates into two major clades largely according to source origin, designated as clades A and B. Clade B strains are community-associated and mostly of non-clinical origin while clade A strains are hospital-associated and mostly of clinical origin (*Galloway-Peña et al., 2012*; *Leavis et al., 2007*; *Palmer et al., 2012*). Clade A has been found to further divide into clade A1, which contains epidemic hospital strains, and clade A2, which encompasses animal strains and strains linked to sporadic human infections (*Lebreton et al., 2013*).

Clade A1 or clonal complex 17 (CC17, a MLST designation) has adapted to the hospital environment and is adept at GIT colonization with the potential to cause invasive disease (*Top, Willems & Bonten, 2008*; *Willems et al., 2005*). Members of clade A1 are characterised by larger genomes and harbour a greater abundance of virulence factors and genes conferring antibiotic resistance compared to non-A1 *E. faecium* lineages, a reflection of adaptation to healthcare environments (*Galloway-Peña et al., 2012*; *Guzman Prieto et al., 2016*; *Top, Willems & Bonten, 2008*).

In Australia, as in other countries, we have observed the sequential emergence of new *E. faecium* clones within the clade A1 hospital lineage which spread rapidly and displace previously endemic clones. For example, from 1994 to 2005, Australian hospital acquired *E. faecium* VRE was uncommon and mostly caused by ST17 strains. The situation changed suddenly from 2005 when there was a nationwide wave of by *E. faecium* ST203 blood stream infections (BSI), a significant and rising proportion of which are *vanB* VRE (*Coombs et al., 2014*; *Johnson et al., 2010*; *Lam et al., 2012*). Previous work comparing ST17 and ST203 genomes revealed that ST203 possesses 40 unique genes with inferred functions of riboflavin metabolism, ion transport and phosphorylation, and harboured a larger vancomycin resistance-conferring Tn*1549* transposon (*Lam et al., 2013*).

At the Austin Hospital in Melbourne, improved cleaning protocols following our local ST203 outbreak were associated with a reduction in VRE BSI between 2009 and 2011 (*Grabsch et al., 2012*). However, despite retaining these protocols we once again observed an abrupt increase in *vanB* VRE *E. faecium* BSI from 2012 onwards that was caused by a completely new ST. We originally recognised the change in strain using PFGE and a high-resolution melt method (*Tong et al., 2011*) but have now switched to whole genome sequencing for epidemiological typing. We lodged the alleles of the new ST with the MLST Database and received the new designation ST796 in September 2012 (*Mahony et al., 2014*). ST796 was unknown before 2011 but by 2013 *vanB* ST796 *E. faecium* had caused a large outbreak of colonisation in a Melbourne Neonatal Intensive Care (*Lister et al., 2015*) and in

the same year was responsible for 40% of *E. faecium* VRE BSI in 5 geographically separate Melbourne hospitals, largely replacing its ST203 predecessor strains. In 2015, ST796 *vanB E. faecium* was responsible for 62 of 117 (53%) of all patient episodes of all *E. faecium* bacteraemia in Melbourne Hospitals, compared with 10 of 117 (8.5%) for ST203.

In the current study, we used single molecule real-time sequencing to establish a high quality, fully assembled genome sequence of ST796 *E. faecium* isolate Ef_aus0233, a representative of this emerging clone and then employed population based comparative genomics to better understand the genetic changes that have accompanied the emergence.

## METHODS

### Bacterial strains
A list of the isolates examined in the study is provided (Table S1). *E. faecium* were cultured as previously described (*Johnson et al., 2010*). Four isolates were randomly selected and sequenced twice, included as technical replicates to ensure analytical reproducibility.

### Whole genome sequencing
Short fragment DNA libraries were generated using the Illumina NexteraXT DNA preparation kit and fragment sequencing was undertaken with the Illumina NextSeq 500 platform using $2 \times 150$ bp chemistry. Highly intact and high quality genomic DNA was extracted from Ef_aus0233 and subjected to Pacific Biosciences SMRT sequencing according to the manufacturer's instructions and sequenced with two SMRT cells on the RS II platform (Pacific Biosciences) using P5-C3 chemistry. Genome assembly was performed using the SMRT Analysis System v2.3.0.140936 (Pacific Biosciences). Raw sequence data were *de novo* assembled using the HGAP v3 protocol with a genome size of 3 Mb. Polished contigs were error corrected using Quiver v1. The resulting assembly was then checked using BridgeMapper v1 in the SMRT Analysis System, and the consensus sequence corrected with short-read Illumina data, using the program Snippy (https://github.com/tseemann/snippy). The final chromosome assembly was validated by reference to a high-resolution NcoI optical map using MapSolver (version 3.10; OpGen, Gaithersburg, MD, USA). Common bacterial DNA base modifications and methyltransferase motifs were assessed using the protocol, RS_Modification_and_Motif_Analysis in the SMRT Analysis System v2.3.0.140936 (Pacific Biosciences).

### Plasmid copy number
The approximate number of plasmid copies per cell for the Ef_aus0233 genome was inferred using differences in Illumina sequence read depth. The read depth of plasmid sequences was compared to the average chromosomal coverage to estimate copy number multiplicity.

### Comparison of completed genomes
Artemis Comparison Tool (*Carver et al., 2005*) was used to align the chromosomes of four fully assembled *E. faecium* genomes. The Ef_aus0233 chromosome was compared against other fully assembled *E. faecium* chromosomes using BLASTn DNA:DNA

comparisons that were undertaken and visualized using Blast Ring Image Generator (*Alikhan et al., 2011*).

### *De novo* assembly and genome annotation

Illumina sequence reads were *de novo* assembled into contigs using Spades v3.6.1 (*Nurk et al., 2013*). The closed Ef_aus0233 genome and Spades contigs were annotated with Prokka (v1.12b) (*Seemann, 2014*) using the Enterococcus database (https://github.com/tseemann/prokka/blob/master/db/genus/Enterococcus) as well as manually annotated protein files derived from two fully assembled *E. faecium* genomes (*Lam et al., 2012*; *Lam et al., 2013*). Multilocus sequence types (STs) were determined using an *in silico* tool (https://github.com/tseemann/mlst). CRISPR databases were used to search for CRISPR sequences (http://crispi.genouest.org and http://crispr.u-psud.fr/Server/) (accessed 19th of May 2016). Sequence files were uploaded to the web based ISsaga (*Varani et al., 2011*)(accessed 11th of February 2016) to detect both the abundance and diversity of insertion elements. Phage discovery was undertaken using the web based resource PHAST (accessed 15th of February 2016) (*Zhou et al., 2011*).

### Variant detection and Bayesian population clustering

Snippy was used to map short read data against the full-assembled Ef_aus0233 genome to call core genome single nucleotide polymorphism (SNP) differences. Hierarchical Bayesian clustering was performed upon a core SNP alignment to assign genomes into discrete populations using hierBAPS with BAPS6 (a prior of 10 depth levels and a maximum of 20 clusters were specified)(*Cheng et al., 2013*). Nested clustering analyses were undertaken upon subsets of the original SNP alignment to a total depth of three levels or until no further clustering could be achieved.

### Recombination and phylogenomic analysis

Recombination within the core genome was inferred using ClonalFrameML v1.7 (*Didelot & Wilson, 2015*) using the whole genome alignment generated by Snippy. The ML tree generated with FastTree v2.1.8 was used as a guide tree for ClonalFrameML. Positions in the reference genome that were not present in at least one genome (non-core) were omitted from the analysis using the "ignore_incomplete_sites true" option and providing ClonalFrameML with a list of all non-core positions. Maximum likelihood trees with bootstrap support were constructed using a recombination free SNP alignment with the program FastTree (*Price, Dehal & Arkin, 2010*). Bootstrap support was derived from comparisons between the original tree against 1,000 trees that were built upon pseudo-alignments (sampled from the original alignment with replacement).

### Pan genome analysis

Orthologous proteins were identified through reciprocal blast using Proteinortho5 v5.11 (*Lechner et al., 2011*). A blast cutoff of 95% identity and alignment coverage of 30% were used. The resulting matrix of ortholog presence and absence was visualized using Fripan (https://github.com/drpowell/FriPan) (downloaded on the 28th of April 2016). The General Feature Format files have been deposited in Figshare

**Table 1  Characteristics of *E. faecium* ST796 Ef_aus0233 complete genome.**

|  | Length (bp) | % G + C | Copy number | No of CDS | No of non-parologous CDS | No of tRNA | No of rRNA |
|---|---|---|---|---|---|---|---|
| Chromosome | 2,888,087 | 38.2 | 1 | 2,726 | 2,644 | 70 | 6 |
| Ef_aus0233_p1 | 197,153 | 35.4 | 1 | 210 | 197 | – | – |
| Ef_aus0233_p2 | 79,293 | 33.8 | 1 | 96 | 96 | – | – |
| Ef_aus0233_p3 | 77,977 | 35.2 | 1 | 84 | 84 | – | – |
| Ef_aus0233_p4 | 22,080 | 35.6 | 2 | 28 | 23 | – | – |
| Ef_aus0233_p5 | 7,837 | 33.5 | 8 | 8 | 8 | – | – |

(https://figshare.com/articles/Evolutionary_origins_of_the_emergent_ST796_clone_of_vancomycin_resistant_Enterococcus_faecium/4007760).

## Sequence alignment and visualization

The alignment of homologous sequences was undertaken using Mauve (*Darling et al., 2004*). Alignments were performed using MUSCLE (*Edgar, 2004*). Sequences and alignments were visualized using Geneious Pro (version 8.1.8, Biomatters Ltd. (www.geneious.com)).

## RESULTS AND DISCUSSION

### Genome overview

Assembly of the 158,885 sequence reads from PacBio SMRT sequencing of Ef_aus0233 (N50 read length 8,952 bp) resulted in reconstruction of a 3,272,427 bp genome, comprising a circular chromosome and five circular plasmids (Table 1). Remaining small hompolymer insertion errors were corrected using Illumina reads. The structural integrity of the chromosome assembly was confirmed correct by reference to a *NcoI* optical map (Fig. 1A). DNA base modification analysis indicated an absence of adenine methylation.

### Antimicrobial resistance gene content

One of the major drivers behind the success of *E. faecium* in the clinical environment is its ability to acquire genes conferring antibiotic resistance (*Handwerger et al., 1993*; *Iwen et al., 1997*; *Murray, 2000*). *In silico* antibiotic resistance screening of the Ef_aus0233 genome confirmed the presence of seven loci, conferring resistance to many major classes of antibiotics including trimethoprim and vancomycin (Table 2). Vancomycin resistance in Ef_aus0233 is conferred by the Tn*1549* transposon, harbouring the *vanB* operon that has integrated into the chromosome. Here, Tn*1549* was 57 Kb in size and inserted into a signal peptidase 1 gene, which is the larger of the two reported versions and is an insertion site that was previously reported in a comparative analysis of ST203 genomes (*Howden et al., 2013*; *Lam et al., 2013*). Of particular interest, the Tn*1549* transposon in Ef_aus0233 and that of the fully assembled ST203 representative genome (Ef_aus0085) (*Lam et al., 2013*), shared 100% pairwise nucleotide identity across the full length of the element, implying a common Tn*1549* origin for these two clones. The majority of ST796 isolates exhibit vancomycin resistance. However, three ST796 vancomycin susceptible
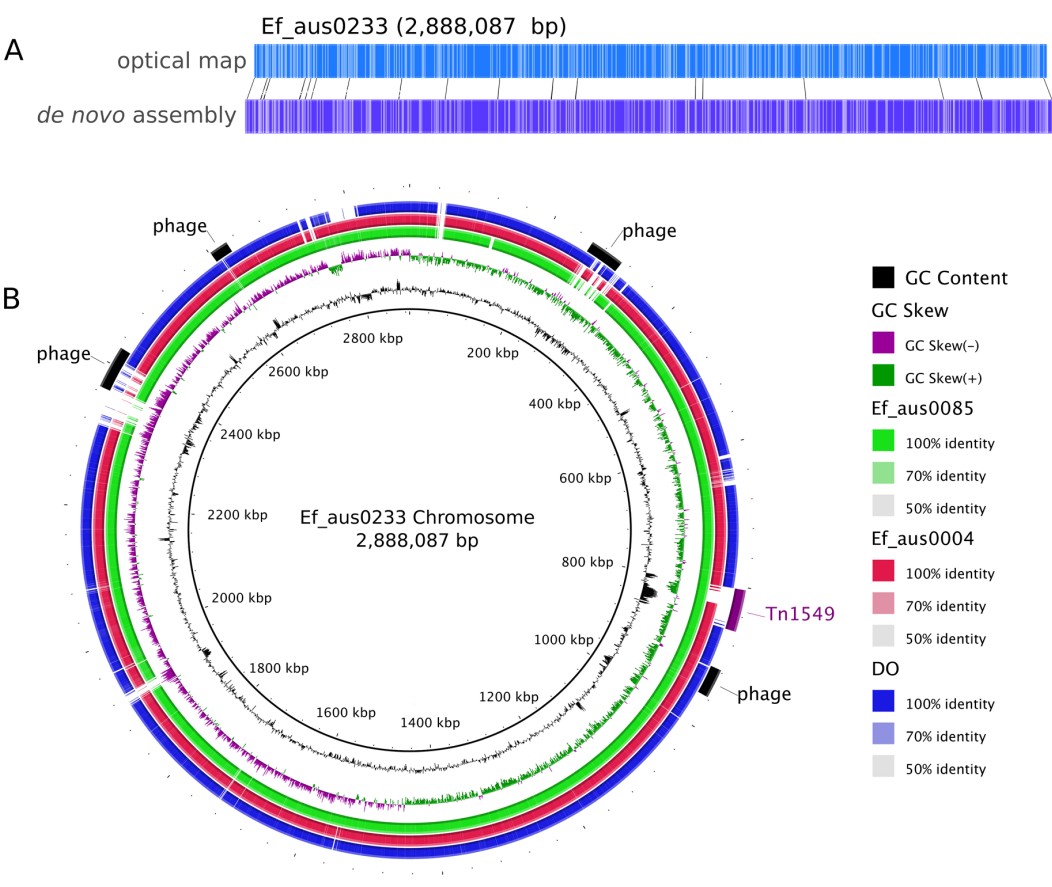

**Figure 1  Ef_aus0233 chromosomal optical map and BRIG plot.** (A) Optical map of the Ef_aus0233 chromosome. (B) Referenced based alignment of blast hits of Ef_aus0085, Ef_aus0004 and DO genomes against the aus0233 chromosome. Prophage elements and the Tn*1549 VanB* containing transposon are annotated as arcs in the outermost ring.

enterococci (VSE) have been isolated to date, one of which (Ef_aus1016) is included in our study and is discussed later.

## Virulence gene content

In addition to antimicrobial resistance genes, virulence related genes are particularly enriched among hospital adapted *E. faecium* strains and are thought to enhance fitness in the hospital environment (*Rice et al., 2003*). *In silico* comparative analysis (see methods) of the Ef_aus0233 genome revealed the presence of several genes associated with virulence including collagen-binding adhesin (*Rice et al., 2003*) (chromosome coordinates 2,235,651–2,233,486), enterococcal surface protein (*Shankar et al., 1999*; *Van Wamel et al., 2007*) (chromosome coordinates 2,786,822–2,780,895), hemolysin (*Cox, Coburn & Gilmore, 2005*) (chromosome coordinates 1,025,987–1,027,363), all of which are also present in the fully assembled genomes of ST17 and ST203 isolates (*Lam et al., 2012*; *Lam et al., 2013*).

The Ef_aus0233 enterococcal surface protein (length = 5,928 nt; 9 rib repeats) shared 83.3% and 82.9% nucleotide identity with the ortholog in Ef_aus0004 (length = 4,938 nt; 6 Rib repeats), a fully assembled ST17 genome (*Lam et al., 2012*) and Ef_aus0085

 

**Table 2  Antibiotic resistance genes and mutations present in Ef_aus0233 and other ST796.**

| Resistance | Product | Gene | Location (nucleotide positions) | Reference |
|---|---|---|---|---|
| Trimethoprim | Dihydrofolate reductase | *dfrG* | 331,475–331,972 (chromosome) | *Sekiguchi et al. (2005)* |
| Tetracycline | Tetracycline resistance protein | *tetM* (Tn*916*) | 652,734–654,653 (chromosome) | *Burdett, Inamine & Rajagopalan (1982)* |
| Macrolides | ABC transporter protein | *msrC* | 2,711,468–2,712,946 (chromosome) | *Portillo et al. (2000)* |
| | rRNA adenine N-6-methyltransferase | *ermB* | 13,080–13,842 (plasmid 4) | *Trieu-Cuot et al. (1990)* |
| Aminoglycosides | Bifunctional aminoglycoside modifying enzyme | *aac*(6′)-*aph2″* | 60,698–62,008 (plasmid 1) | *Patterson & Zervos (1990)* |
| | | | 60,366–61,805 (plasmid 3) | |
| Vancomycin | *VanB* ligase | *vanB* (Tn*1549*) | 803,567–861,054 (chromosome) | *Arthur, Reynolds & Courvalin (1996)* |

(length = 5,199 nt; 7 Rib repeats), a fully assembled ST203 genome (*Lam et al., 2013*), respectively (Fig. S1). The hemolysin of Ef_aus0233 shared complete identity with orthologs in Ef_aus0004 and Ef_aus0085, suggesting that this CDS may be under strong selection.

## Insertion sequence content

The Ef_aus0233 chromosome was found to contain 80 distinct elements (9 families) while Ef_aus0233_p1 had 41 (6 families), Ef_aus0233_p3 had 8 (5 families), Ef_aus0233_p4 had 8 (2 families) and no IS elements were detected on Ef_aus0233_p2 or Ef_aus0233_p5. Several of these IS families have been found not only in enterococci but additionally in species of other genera, including *Carnobacterium* and *Lysinibacillus*, reflecting the easy by which *E. faecium* can acquire exogenous DNA (*Guzman Prieto et al., 2016*).

## CRISPR content

Akin to an adaptive immune system, the clustered regularly interspaced short palindromic repeats (CRISPR) systems of prokaryotes function as a sequence-specific security to defend genomes against viral predation and exposure to invading nucleic acid (*Horvath & Barrangou, 2010*). Unlike members of the community-associated *E. faecium* lineage, genomes belonging to the CC17 *E. faecium* have been found to lack CRISPR systems (*Van Schaik et al., 2010*). Given the advantages associated with the acquisition of extraneous DNA carrying antimicrobial resistance genes, CRISPRs are thought to be under negative selection among multi-drug resistant enterococci (*Palmer & Gilmore, 2010*). Despite this, two distinct CRISPR loci were detected on Ef_aus0233_p1 (chromosome coordinates 168197–168396) and Ef_aus0233_p2 (chromosome coordinates 2,630–2,860), both containing three spacers and imperfect direct repeats. A single CRISPR associated gene (cas2) was detected. However, no cas1 ortholog was detected. Due to the practical necessity of cas1 for the operation of CRISPR systems (*Yosef, Goren & Qimron, 2012*), it is unlikely that these detected CRISPR systems are functional.

## Prophage content

The Ef_aus0233 genome was found to contain five putative prophages. Prophages Ef_aus0233_chr_phage-1 (chromosome coordinates 260,208–308,229: 52 CDS), Ef_aus0233_chr_phage-2 (chromosome coordinates 916,578–956,914: 61 CDS), Ef_aus0233_chr_phage-3 (chromosome coordinates 2,366,359–2,425,249: 78 CDS) and Ef_aus0233_chr_phage-4 (chromosome coordinates 2,601,769–2,627,494: 19 CDS) were located on the chromosome (Fig. 1B) while Ef_aus0233_p1_phage-1 (plasmid-1 coordinates 62,059–94,736: 30 CDS) was identified on Ef_aus0233_p1. Alignment of these prophage elements signified that several common blocks of co-linearity existed (Fig. S2); however, an overall lack of prophage genome conservation implies that these phage represent five distinct elements. Prophage gene content among a diverse collection of *E. faecium* genomes is discussed below.

## Comparisons with other completed *E. faecium* genomes

In addition to diversity within the core and accessory genome, structural rearrangements represent an additional layer of genomic variation that may contribute to *E. faecium* phenotypic differences (*Lam et al., 2012*; *Lam et al., 2013*; *Matthews & Maloy, 2010*). To assess how the genomic organization of the Ef_aus0233 chromosome compared to that of other *E. faecium* genomes, a whole chromosome alignment of Ef_aus0233, Ef_aus0004, Ef_aus0085 and DO was undertaken. The BLASTn based alignment revealed substantial conservation of genome content (Fig. 1B) and chromosomal architecture (Fig. 2A). Like Ef_aus0085 and DO, Ef_aus0233 does not exhibit the replichore inversion observed in Ef_aus0004 (*Lam et al., 2012*; *Lam et al., 2013*).

### *E. faecium* population genomic comparisons

In order to contextualize the 21 ST796 genomes within the global diversity of *E. faecium* as a species, we compared these genome data with a diverse collection of 89 published, fully assembled and draft *E. faecium* genomes (Table S1). To investigate the structure and evolutionary relatedness of the strains, we employed an unsupervised Bayesian clustering technique (BAPS) to distinguish distinct genomic populations and estimated a rooted phylogenomic tree using maximum likelihood.

Here, we found that our BAPS groups unambiguously classified genomes into the two previously reported A and B clades (*Lebreton et al., 2013*) (Fig. 3A). BAPS-1 corresponded with clade A and BAPS-2 corresponded with clade B, while BAPS-1, BAPS-1.1-4 and BAPS-1.5 overlapped with clades A1 and A2 (Fig. 3B). All ST796 isolates clustered within clade A1.

When inspecting the phylogeny of the ST796 genomes, it was noted that ST796 and ST555 share a most recent common ancestor (MRCA) (Fig. 3C). Without exception, ST555 and ST796 genomes formed distinct monophyletic clades that were in agreement with the BAPS groupings (BAPS 1.3.3 and 1.3.4, respectively). The ST555 clone is another recently-emerged hospital adapted ST, however its discovery in the hospital environment predates that of ST796 (*Coombs et al., 2014*). Another major difference between these two STs is that ST796 appears to have been localized to south east Australia and New Zealand

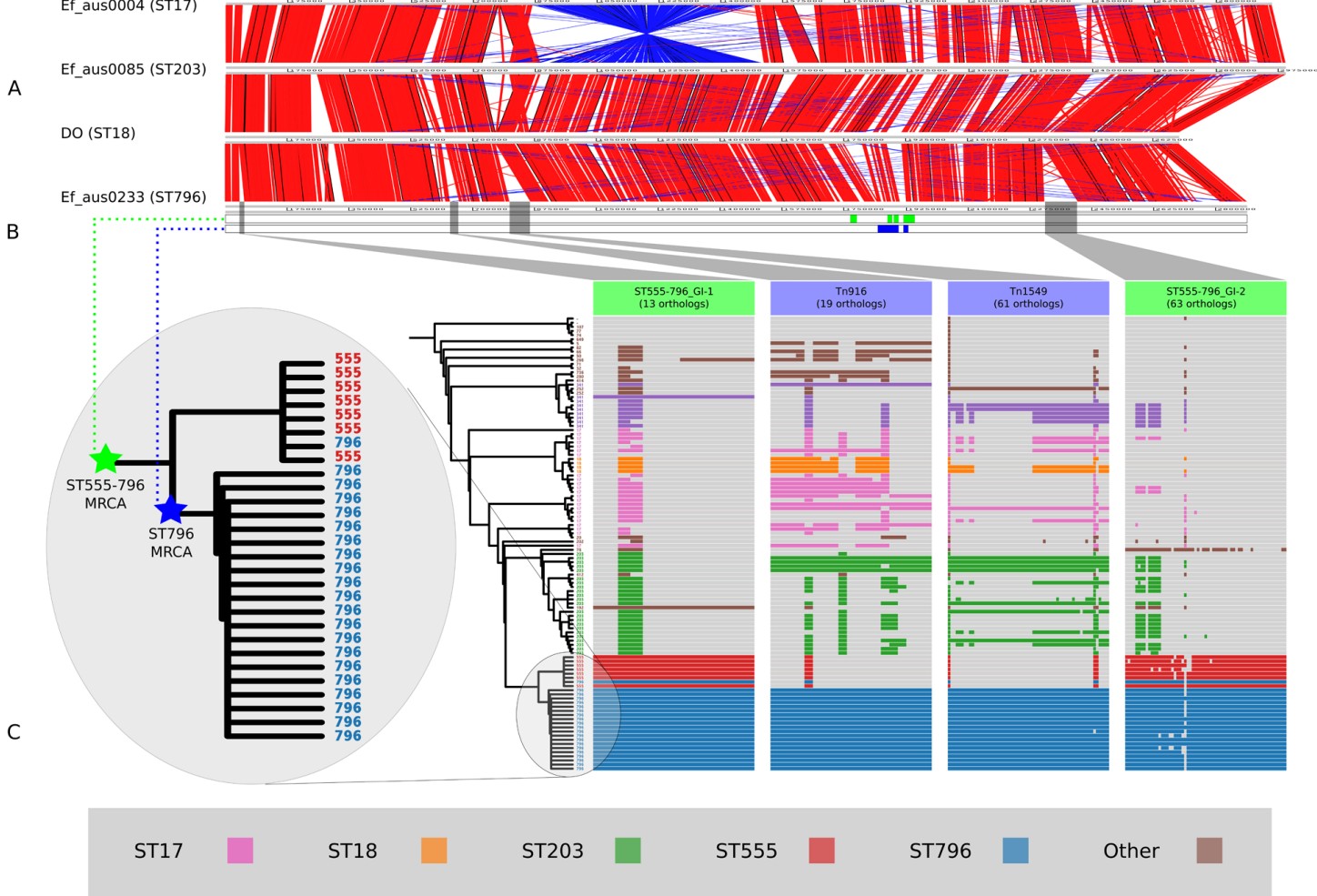

**Figure 2 Comparisons of chromosomal architecture, genomic islands and recombining segments associated with the ST555-796 and ST796 clades.** (A) Alignment of fully assembled chromosomes of Ef_aus0233, Ef_aus0085, Ef_aus0004 and DO. (B) Conservation of recombining segments in the ST555-796 and ST796 MRCAs. (C) Core genome phylogeny aligned with gene content blocks for identified genomic islands. Colours indicate the MLST designations.

(*Carter et al., 2016*; *Coombs et al., 2014*), while ST555 has been reported nationally in the Northern Territory, South Australia and Western Australia (*Coombs et al., 2014*), in China (*Liu et al., 2011*) and among wild birds in the United States (*Oravcova et al., 2014*). The national and international pervasiveness of ST555 and relatively limited geographical dispersal of ST796 in southeast Australia is consistent with a scenario in which the evolutionary emergence of ST555 predates that of ST796.

Recombination analyses indicated that both the ST555-796 and ST796 MRCAs have evolved in part by recombination. Inspection of the inferred recombining segments for these two ancestors revealed a single hotspot of 170 kb that contained two overlapping clusters of increased SNP density (ST555-796_MRCA: chromosome coordinates 1,783,249–1,953,029, ST796_MRCA: chromosome coordinates 1,857,926–1,937,284) (Fig. 2B). The spatial clustering of these inferred ancestral recombination events suggests that this region

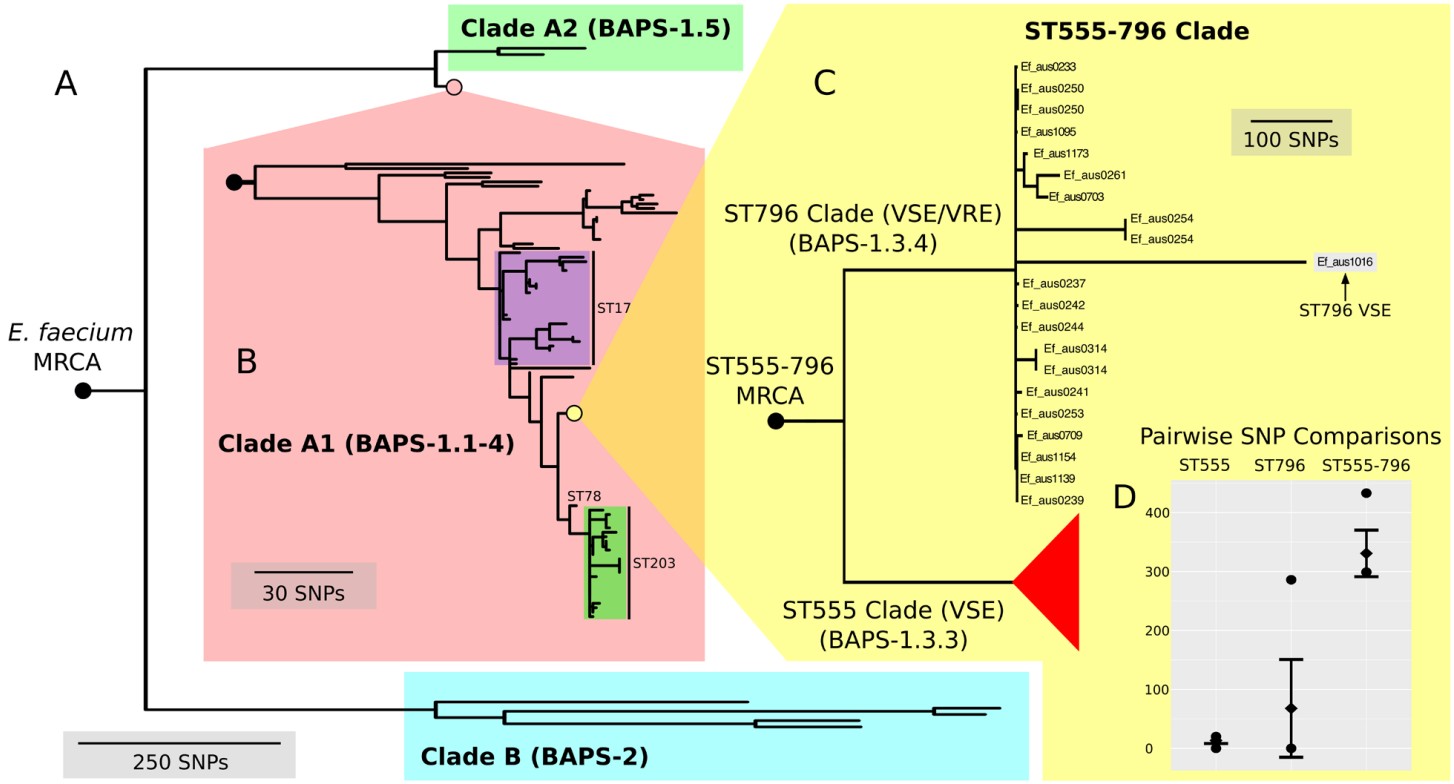

**Figure 3** **Nested core genome SNP phylogeny of the greater *E. faecium* population.** (A) *E. faecium* population tree containing the major division between the community and hospital associated clades (B and A). (B) Nested tree focusing on the A1 clade containing the hospital associated isolates. (C) Nested tree focusing on the ST555-796 containing clade. Nodes with less than 70% bootstrap support have been collapsed. The VSE ST796 isolate is located within the ST796 clade. (D) Pairwise core SNP comparisons of within and between group diversity for the ST555 and ST796 clades (ST555: 13 SNPs, ST796: 68 SNPs, ST555-796: 330 SNPs). *Y*-axis depicts the number of SNP differences, error bars indicate one standard deviation above and below the mean and points represent the minimum and maximum values.

may contain non-clonally derived alleles (particularly those in the ST796 MRCA) that may have been under positive selection and perhaps contributed to the emergence of ST796. Inspection of this region revealed a preponderance of cell-wall associated transport CDS, including CDS encoding putative copper and cadmium-translocating P-type ATPases, amino acid permeases, OxaA-like membrane protein, as well as housekeeping CDS such as Glycyl-tRNA synthetase subunits, RecO DNA repair proteins and a GTP-binding protein.

In order to assess the evolutionary divergence between the ST555 and ST796 clades, a ST555-796 specific core genome was established and pairwise SNP distances were calculated. As we were primarily interested in SNPs derived through clonal evolution, again we removed SNPs inferred to have arisen through recombination. Inspection of SNP distribution in VSE Ef_aus1016 revealed several dense clusters, indicating substantial recombination, which was subsequently removed by a second round of recombination detection. Consistent with two distinct groups, within clade comparisons revealed smaller mean SNP differences (within-ST555: 13 SNPs, within-ST796: 68 SNPs) than that between clades (between ST555 and ST796: 330 SNPs) (Fig. 3D).

## Ancestral single nucleotide polymorphisms

Forty-one core-genome SNPs differentiated the ST555–ST796 MRCA from its predecessors, while only two core-genome SNPs were predicted in the ST796 MRCA compared to ST555 (Table S2). Analysis of these SNPs showed a range of CDS impacted. Among the 41 SNPs, 22 were predicted to change amino-acid sequence and potentially alter protein function, including non-synonymous mutations in four CDS encoding putative regulatory proteins. While the function of these regulatory proteins and the consequence of the predicted mutations is unknown, such changes can have profound impacts on phenotype (*Howden et al., 2011*; *Howden et al., 2008*).

## Accessory gene content comparisons

The clustering of all predicted CDSs into orthologous groups allowed for inter-ST comparisons at the gene-content level. In total, there were 10,740 orthologous clusters among the 110 genomes, of which 1,437 were core and 9,303 were variably present (accessory)—representing the *E. faecium* pan genome (Table S3). Using this approach, orthologous clusters that were diagnostic of the ST555–ST796 and ST796 populations were identified.

## Lineage specific genomic islands

The acquisition of genomic islands has been reported in previous studies that compared the genomes of hospitalized and non-hospital derived isolates, suggesting that such novel elements may offer possessing strains a competitive advantage (*Heikens et al., 2008*). In this study, subsets of the accessory genome that were associated with ST555–ST796 and ST796 genomes were found to cluster on the Ef_aus0233 chromosome (Fig. 2C). The contiguous location of these CDS and conserved inheritance patterns, indicated that these elements collectively formed larger genomic islands and were likely to have been acquired through horizontal gene transfer events. Given the ancestral relationships among the genomes and the conservation of ortholog presence amongst these lineages, it is reasonable to infer that these events occurred at various stages along the evolutionary paths of the ST555–ST796 and ST796 ancestries.

Two genomic islands were conserved among ST555–ST796 genomes while being almost entirely absent from genomes of other STs (ST555–ST796_GI-1, chromosome coordinates 39,093–53,122: 13 CDS, and ST555–ST796_GI-2 chromosome coordinates 2,316,643–2,373,309: 63 CDS), suggesting that these elements are likely to have been acquired by the ST555–ST796 MRCA. Assessment of the CDS annotation for ST555–ST796_GI-1 (56 kb) suggests it is a mosaic integrative element. Two 3 kb regions spanning a site-specific tyrosine recombinase and excisonase at the 5′ end of this element and replication proteins at the 3′ extreme were identical to a previously described *Enterococcus faecalis* pathogenicity island (*Shankar, Baghdayan & Gilmore, 2002*). A 13 kb region harbouring a putative beta-galactosidase and other sugar modifying CDS was identical to a region in ST203 Ef_aus0085 (*Lam et al., 2013*). The function of CDS in the remaining 33 kb was more difficult to infer with few database matches to indicate function. However, a role for this region in cell wall modification (potentially DNA transfer) was suggested by the presence
of CDS encoding cell-wall binding proteins, peptidases, ATP/GTP-binding proteins and peptidoglycan-binding proteins. ST555–ST796_GI-2 (14 kb) is another integrative element with CDS encoding a site-specific tyrosine recombinase, replication proteins, and putative sugar kinases, hydrolases and permeases. This potential carbohydrate utilization/transport locus shared complete nucleotide sequence identity with a region of the *Enterococcus gallinarum* genome (strain ID: FDA ARGOS_163, NCBI BioProject: PRJNA231221). The phage CDS content and their predicted products is provided (Table S4).

Two other elements were found to be exclusively present among ST796 genomes in this comparison. One of these was the Tn*916*-like transposon (chromosome coordinates 635,179–658,601: 19 CDS), carrying tetracycline resistance (*Franke & Clewell, 1981*) and the second was the Tn*1549* transposon (chromosome coordinates 803,567–861,054: 61 CDS), carrying vancomycin resistance (*Garnier et al., 2000*). Given the phylogeny, it appears likely that the ST796 MRCA acquired these two transposons and then spread (Fig. 2C). The exception to this pattern was the single VSE ST796 genome (Ef_aus1016) that lacked these elements that were universally conserved among VRE ST796 genomes. Ef_aus1016 exhibits the same genomic island presence/absence profile that is observed among the ST555 genomes. Given this, it appears that Ef_aus1016 might be an extant descendent of an ST796 evolutionary intermediate that had not yet acquired the ST796 specific GIs, such as Tn*1549*, suggesting that this lineage was originally VSE. The horizontal acquisition of Tn*1549* has been demonstrated (*Launay et al., 2006*) and evidence for a VSE version of an emergent *E. faecium* clone to precede the VRE version has been previously documented with the emergence of ST17, ST203 and ST252 (*Johnson et al., 2010*).

## Prophage gene content comparisons

Alignment of the orthologs found within the prophages that were identified in the Ef_aus0233 genome revealed the extent to which these elements are conserved among the greater *E. faecium* population (Fig. 4). The majority of orthologs within these prophages were found to exist in non-ST796 genomes, suggesting that at the gene-content level, such prophages are not unique to ST796. Prophages Ef_aus0233_chr_phage-2, Ef_aus0233_chr_phage-4 and Ef_aus0233_p1_phage-1 showed the greatest degree of ortholog conservation with non-ST796 genomes. Prophages Ef_aus0233_chr_phage-1 and Ef_aus0233_chr_phage-3 did contain orthologs present in non-ST796 genomes, however the presence of these orthologs outside of ST796 genomes was limited. Overall the prophages in Ef_aus0233 form a substantial contribution to the accessory genome but do not contain CDS that are unique to ST796 (Table S4).

## Plasmid gene content comparisons

Plasmids form an important component of the *E. faecium* accessory genome that can spread horizontally through a population and carry genetic elements that may confer enhanced fitness (*Fiedler et al., 2016*). Approximately 12% of the Ef_aus0233 genome (384,340 bp) is comprised of plasmid DNA. In order to assess the conservation of plasmid gene content among ST796 genomes and across the greater *E. faecium* population, the presence and absence of plasmid genes within the ortholog clusters were inspected (Fig. 5). Patterns of
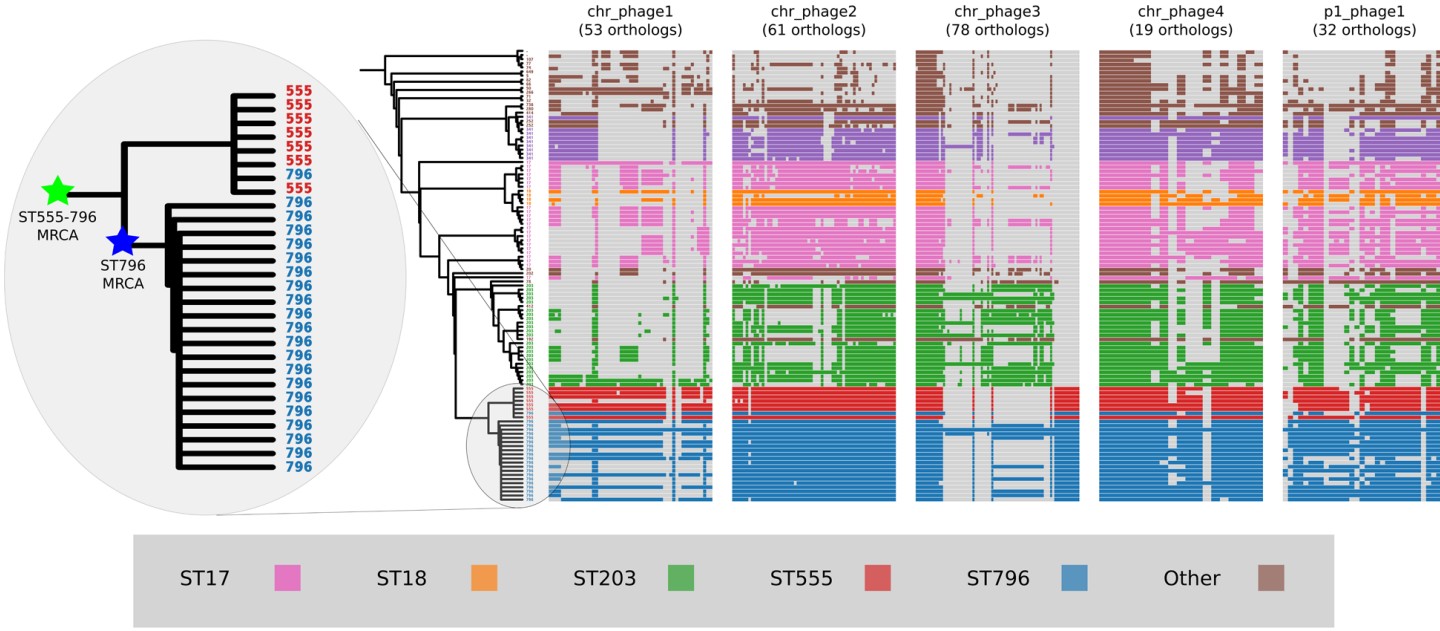

**Figure 4** **Prophage gene content comparisons: the presence and absence of orthologs within each of the five phages that were identified in the Ef_aus0233 genome.** The phylogeny depicts the evolutionary relationships among the genomes. Colours indicate the MLST designations.

individual ortholog presence and absence demonstrated that all plasmid orthologs were found in non-ST796 genomes, however in varying degrees. Plasmids Ef_aus0233_p1, Ef_aus0233_p4 and Ef_aus0233_p5 contain orthologs found outside ST796, however they are rarely seen elsewhere in their entirety. A list of all plasmid CDS content and their predicted products is provided (Table S5).

Given these gene content patterns and the aforementioned phylogenomic relationships between ST555 and ST796 genomes, it appears likely that the ST555–ST796 MRCA acquired these plasmids, as they are not observed in their entirety in surrounding clades. Interestingly, Ef_aus0233_p2 was not only scarce among non-ST796 genomes but lacked conservation among the ST796 genomes. Overall, no single plasmid ortholog was specific to ST796, however given the set of isolates analyzed in this study, such plasmids, in their entirety, appear to be diagnostic of the ST555–ST796 lineage. Furthermore, the intra-ST796 differences in plasmid gene content, particularly in Ef_aus0233_p2, indicate there are appreciable amounts of diversity within the ST796 accessory genome, variation that might be useful during outbreak investigations involving this clone (*Lister et al., 2015*).

## CONCLUSION

The hospital environment presents a challenging ecological niche for the adaptation of bacterial pathogens. Historically, *E. faecalis* was the leading causative agent of enterococcal nosocomial infections, however *E. faecium* infections have escalated in the last decade (*Galloway-Peña et al., 2009*; *Guzman Prieto et al., 2016*; *Leavis, Bonten & Willems, 2006*; *Willems et al., 2011*; *Willems & Van Schaik, 2009*). Following this apparent interspecies replacement, population-based studies have observed substantial intraspecies dynamics

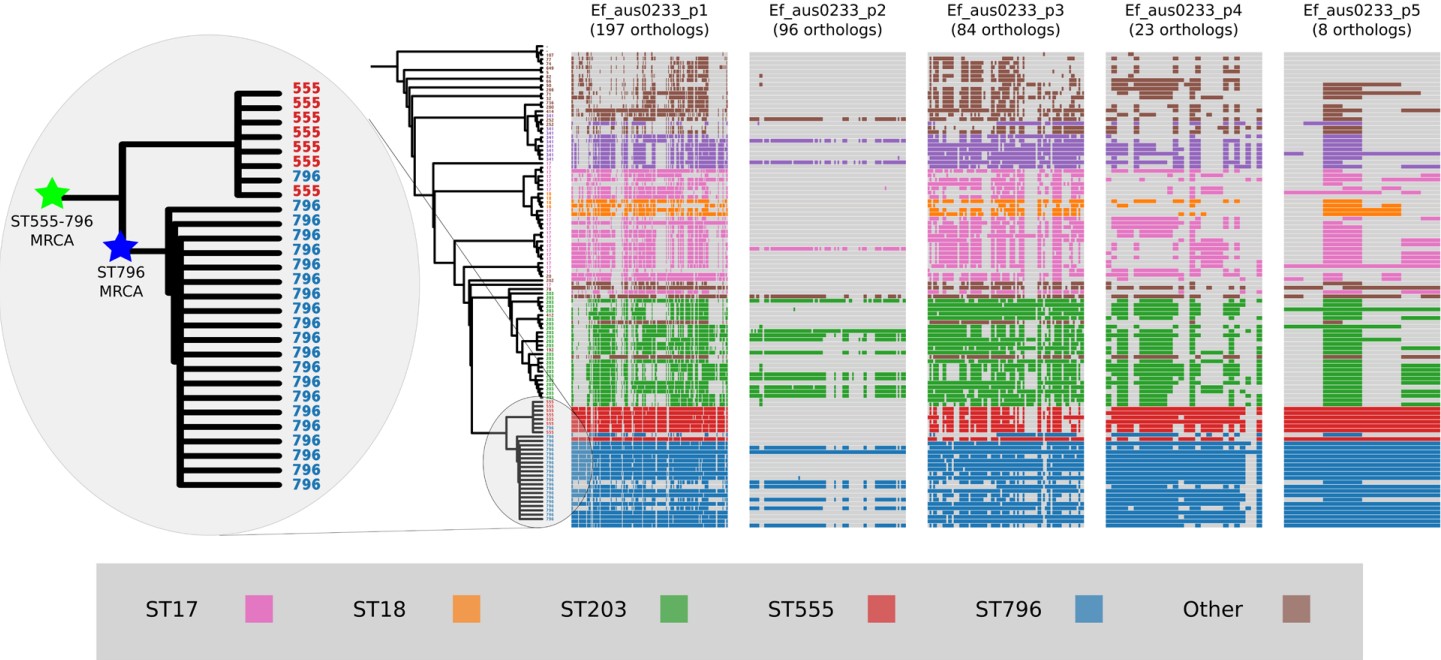

**Figure 5** Plasmid gene content comparisons: the presence and absence of orthologs within each of the five plasmids that were identified in the Ef_aus0233 genome. The phylogeny depicts the evolutionary relationships among the genomes. Colours indicate the MLST designations.

with clonal replacement of *E. faecium* STs in hospitals (*Bender et al., 2016*; *Johnson et al., 2010*). Here we have described the genomic basis for the emergence of a new highly hospital adapted *E. faecium* ST early in its evolutionary history. The preparation of a fully assembled ST796 genome facilitated a comprehensive genomic analysis of this lineage and enabled detailed comparisons among other clinically relevant draft and fully assembled *E. faecium* genomes.

We demonstrate that the emergence of ST796 was preceded by several genomic events including the acquisition of two genomic islands, plasmid and phage activity, modest SNV accumulation and recombination. These analyses highlight genetic elements within the *E. faecium* core and accessory genome that may have been important drivers for the evolution of the ST555–ST796 and ST796 lineages. Given the likely significance of genomic island acquisition for the emergence of CC17 (*Heikens et al., 2008*), the GIs presented in this study presumably reflect adaptive responses to the clinical environment, either through acquired antibiotic resistance or perhaps enhanced capacity to utilize carbohydrates and thus augment gastrointestinal colonization. Our finding that ST796 shares a MRCA with ST555 and may have emerged from a VSE evolutionary intermediate is another example of newly emergent VRE arising from a VSE MRCA, although ST796 itself is almost exclusively VRE when identified in human BSIs unlike ST555 which causes both VSE and VRE BSI in Australian hospitals in about equal proportions (*Coombs et al., 2014*).

This analysis focused upon providing an overview of the first fully assembled ST796 genome and genomic differences that were assessed at the inter-ST population level. In order to explore specific diversity within the ST796 lineage, an intra-ST population study

focusing upon diversity among a large collection ST796 genomes is currently underway. Our observation of substantial variation within the ST796 accessory genome, in particular plasmid presence and absence, suggest a means for effective intra-ST796 genotyping that could potentially be more useful than core genome analysis in the tracking of outbreaks.

Hospitals are controlled but nonetheless dynamic environments. Examining pathogen genomic changes in these environments is important to understanding the bacterial response to human interventions, such as changing infection control or antibiotic stewardship practices. In this study we deconstructed the genomic events that have shaped the evolution of a highly successful *E. faecium* clone. With our current state of knowledge, we know that there is rapid genomic change occurring however we don't fully understand the consequences of such changes, as a large percentage of identified *E. faecium* genes have unknown functions. The promise of genomics will only be realized when we can combine our insights on genomic evolution with the functional consequences of such changes on pathogen phenotypes. The research presented here incrementally builds our understanding and provides a solid basis for future studies, as both clinical and public health microbiology transition into the genomic era.

### Funding

This research was supported by the National Health and Medical Research Council (NHMRC) of Australia (1027874, 1084015). BPH and TPS are recipients of NHMRC fellowships (1023526 and 1008549, respectively). SG and TS was supported in part by the VLSCI Life Sciences Computation Centre, a collaboration between Melbourne, Monash and La Trobe Universities, and an initiative of the Victorian Government, Australia. AAM is a recipient of an Australian Postgraduate Award, University of Melbourne. The funders had no role in study design, data collection and analysis, decision to publish, or preparation of the manuscript.

### Grant Disclosures

The following grant information was disclosed by the authors:
National Health and Medical Research Council (NHMRC) of Australia: 1027874, 1084015.
NHMRC fellowships: 1023526, 1008549.
VLSCI Life Sciences Computation Centre.
Australian Postgraduate Award.

### Competing Interests

The authors declare there are no competing interests.

### Author Contributions

- Andrew H. Buultjens and Timothy P. Stinear conceived and designed the experiments, performed the experiments, analyzed the data, wrote the paper, prepared figures and/or tables, reviewed drafts of the paper.

- Margaret M.C. Lam conceived and designed the experiments, performed the experiments, analyzed the data, reviewed drafts of the paper.
- Susan Ballard, Elizabeth A. Grabsch and Torsten Seemann performed the experiments, contributed reagents/materials/analysis tools, reviewed drafts of the paper.
- Ian R. Monk performed the experiments, analyzed the data, reviewed drafts of the paper.
- Andrew A. Mahony, M. Lindsay Grayson, Geoffrey W. Coombs and J. Owen Robinson contributed reagents/materials/analysis tools, reviewed drafts of the paper.
- Stanley Pang contributed reagents/materials/analysis tools.
- Paul D.R. Johnson conceived and designed the experiments, analyzed the data, wrote the paper, reviewed drafts of the paper.
- Benjamin P. Howden conceived and designed the experiments, analyzed the data, reviewed drafts of the paper.

## DNA Deposition

The following information was supplied regarding the deposition of DNA sequences:
   DNA sequence data deposited in ENA and GenBank:
   https://www.ncbi.nlm.nih.gov/bioproject/PRJEB14733/.

## Data Availability

   Buultjens, Andrew (2016): Evolutionary origins of the emergent ST796 clone of vancomycin resistant Enterococcus faecium. figshare.
   https://dx.doi.org/10.6084/m9.figshare.4007760.v1.

## Supplemental Information

Supplemental information for this article can be found online at http://dx.doi.org/10.7717/peerj.2916#supplemental-information.

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
