# Peer review of "Evolutionary origins of the emergent ST796 clone of vancomycin resistant Enterococcus faecium"

_PeerJ, doi:10.7717/peerj.2916_

## Round 0.1 · original submission · Minor Revisions

As you can see both reviewers are enthusiastic and have recommended publication of your manuscript. Both have suggested a set of minor improvements. I would be grateful if you could respond in full to their suggestions and update the manuscript where indicated. In particular please ensure that the datasets and supplementary information are accessible before resubmission.

·

Basic reporting

Some minor issues (further specified below).

Experimental design

No comments

Validity of the findings

The conclusions of the paper are supported by the data.

Additional comments

The manuscript by Buultjens et al. describes a genome sequence analysis of an isolate from a recently emerged E. faecium lineage (ST796) that has emerged a nosocomial clone in Oceania. The manuscript provides interesting information on the evolution of this lineage and provides insights into the genes that may contribute to the success of ST796 in the hospital environment. My comments below are intended as suggestions for the authors to further improve their manuscript.

l. 44. It would be more appropriate to refer to Homan et al., 2002 (PMID: 12037049) as this is the study in which the E. faecium MLST scheme was introduced.
l. 119. Explain the abbreviation BRIG (BLAST Ring Image Generator).
l. 173: I believe ‘their’ should be ‘its’
l. 179: It is unclear what is meant with ‘two reported versions’. I believe this refers to the authors’ previous study (Howden et al., 2013 in mBio)? Please expand and/or add references.
l. 194, Cox et al., 2005 describe cytolysin, a protein that is produced by some strains of E. faecalis. Cytolysin is not produced by E. faecium so it is unclear what hemolysin the authors refer to here.
l. 199 – 201: The line that starts with ‘In comparison …’ is incomplete so it is not entirely clear what point the authors want to make. They could consider adding a figure with different versions of the esp gene to (the supplementary materials of) the manuscript.
l. 202 - 205. It is unsurprising that genes encoding gelatinase, aggregation substance and hemagglutinin are not found in this strain, as these virulence factors are specific to E. faecalis (articles reporting their presence in E. faecium generally use flawed methods for species determination in Enterococcus). Please add the information that these genes are found only in E. faecalis or delete these lines.
l. 212. Write ‘the easy by which’
l. 298. It is interesting that the authors describe potential recombination events that were not detected by ClonalFrameML. Were these sites identified by other algorithms for the detection of recombination in microbial genomes (e.g. Gubbins or BratNextGen).
l. 401. The line ‘Overall, no single plasmid ortholog was specific to ST796, however these plasmids in their entirety are diagnostic of the ST555-ST796 lineage.’ is somewhat ambiguous. Do the authors mean to say that the combination of these plasmids is only found in the ST555-ST796 lineage? Anyway, it is tricky to state that these plasmids can be ‘diagnostic’ of this lineage, as it is likely that there are strains out there that have lost one or more plasmids, something which cannot be ruled out on the basis of the limited geographic sampling and relatively low number of strains in this study.
Table 2: msrC and aac(6')- aph2' should be written in italics.
Fig. 2. This figure is somewhat confusing, possibly because of the placement of the panel labels, which do not seem to match with the figure legend. Panel B may be better described as ‘Conservation of recombining segments in the ST555-796 and ST796 MRCAs.’ to highlight that this figure does not show recombination, but the variable presence of the recombining among the different E. faecium isolates
l. 695. Write E. faecium in italics.
Table S1: labels columns L-O with BAPS (not BAP)

Reviewer 2 ·

Basic reporting

No Comments.

Experimental design

No Comments.

Validity of the findings

Minor comments:
1. ENA or SRA identifier for the deposition of raw reads for the Illumina and PacBio sequencing were not provided, as would be expected for this sort of study.
2. NCBI BioProject ID for genome project not provided, as I would have expected.
3. The two Figshare addresses did not work so I could not look up those figures.

Additional comments

In this thorough, well designed and executed study, the authors sequenced an ST796 strain of Enterococcus faecium using Illumina short-read and PacBio long-read technologies, inferred the full genome sequence of the chromosome and associated plasmids, verified it with optical mapping, and performed comparative genomic analyses in order to comment on events in the genome evolution of this isolate. The aims of the study are stated clearly. The analyses are technically sound, using standard, appropriate methods. The text, methods, analyses, figures and tables are clear, appropriate, concise and easy to follow.

I am recommending that this study be accepted for publication without amendments, but I would encourage the authors to re-read the final paragraph of the discussion (L442-445) and perhaps speculate on how this study will inform future studies. Stated in another way, the fundamental issue with this, and similar studies, is that we already know that bacteria that are successful in a hospital environment have experienced genomic changes involving lateral gene transfer, recombination, and genomic rearrangements. Describing these changes in detail, in a particular isolate or outbreak is useful, but what does it really tell us? How does it help us treat, contain or prevent future outbreaks? How can this information help us understand or predict general principles of bacterial evolution in hospital environments? How can we use this information to understand what order these evolutionary events occurred in, and thus, at what point could these events be detected (if at all) and used to prevent the final genomic events that lead to a new, very successful lineage? Did all these genomic events happen almost at the same time or did they evolve sequentially over several generations? What bacterial "species" did the acquired DNA come from? Where in the host or the environment are the reservoirs for the acquired DNA? What would the perfect data set look like that could start to answer these sorts of questions?

Very minor points:
1. GIT acronym is used before first described.
2. In Table 1, a footnote could be added to indicate whether "-" means zero, data not known or not relevant.
3. When using Prokka, the quality of the annotation would depend on the references in the sequence database. Could the authors list the other E. faecium genomes in the database used?

---

## Round 0.2 · accepted · Accept

Thank you for your considered responses to each of the reviewer comments and the improved manuscript. I now consider the manuscript ready for publication. Thank you for sending this exciting work to PeerJ!